# Imaging of Virus-Infected Cells with Soft X-ray Tomography

**DOI:** 10.3390/v13112109

**Published:** 2021-10-20

**Authors:** Damià Garriga, Francisco Javier Chichón, Bárbara M. Calisto, Diego S. Ferrero, Pablo Gastaminza, Eva Pereiro, Ana Joaquina Pérez-Berna

**Affiliations:** 1ALBA Synchrotron Light Source, 08290 Cerdanyola del Vallès, Spain; dgarriga@cells.es (D.G.); bcalisto@cells.es (B.M.C.); epereiro@cells.es (E.P.); 2Centro Nacional de Biotecnología, Consejo Superior de Investigaciones Científicas, 28049 Madrid, Spain; fjchichon@cnb.csic.es (F.J.C.); pgastaminza@cnb.csic.es (P.G.); 3Institut de Biologia Molecular de Barcelona, Consejo Superior de Investigaciones Científicas, Parc Científic de Barcelona, 08028 Barcelona, Spain; dfecri@ibmb.csic.es

**Keywords:** cryo–soft X-ray tomography (cryo-SXT), hepatitis C virus (HCV), vaccinia virus (VACV), Zika virus (ZIKV), direct-acting antiviral (DAAs)

## Abstract

Viruses are obligate parasites that depend on a host cell for replication and survival. Consequently, to fully understand the viral processes involved in infection and replication, it is fundamental to study them in the cellular context. Often, viral infections induce significant changes in the subcellular organization of the host cell due to the formation of viral factories, alteration of cell cytoskeleton and/or budding of newly formed particles. Accurate 3D mapping of organelle reorganization in infected cells can thus provide valuable information for both basic virus research and antiviral drug development. Among the available techniques for 3D cell imaging, cryo–soft X-ray tomography stands out for its large depth of view (allowing for 10 µm thick biological samples to be imaged without further thinning), its resolution (about 50 nm for tomographies, sufficient to detect viral particles), the minimal requirements for sample manipulation (can be used on frozen, unfixed and unstained whole cells) and the potential to be combined with other techniques (i.e., correlative fluorescence microscopy). In this review we describe the fundamentals of cryo–soft X-ray tomography, its sample requirements, its advantages and its limitations. To highlight the potential of this technique, examples of virus research performed at BL09-MISTRAL beamline in ALBA synchrotron are also presented.

## 1. Introduction

Viruses occur universally, presumably infecting all cellular life from the early stages of life development on the planet [1]. For humans, viruses represent a major burden, causing dramatic losses of human life worldwide, and constitute a challenge for society, public health and the economy, as exemplified by the current SARS-CoV-2 pandemic.

Despite their vast diversity, viruses share some common features, the fundamental one being that all viruses are obligate parasites, lacking metabolic mechanisms of their own to generate energy or to synthesize proteins. Due to their unavoidable interaction with the cellular environment, any proliferative virus infection of a host cell leads, to a larger or smaller degree, to alterations in the cell metabolism and/or its substructures, to build a favorable environment for viral multiplication [2,3].

In eukaryotic viruses, one of the most relevant of such alterations is the formation of viral factories or viral replication compartments. These are complex, dynamic, intracellular compartments or inclusions that arise from extensive rearrangement of host cell cytoskeletal and membrane compartments and are used by the virus as platforms for genome replication and morphogenesis. Viral factories (VFs) increase the efficiency of viral replication by concentrating the required cellular and viral materials while shielding the virus from host defenses [4,5,6,7].

VFs come in a myriad of shapes, sizes and locations, depending on the virus that produces them and the stage of the infection cycle. For instance, for some DNA viruses that replicate inside the nucleus, such as Baculoviridae, Herpesviridae, Alloherpesviridae and Polyomaviridae, the formation of subnuclear replication compartments has been reported [8,9,10,11,12]. Regarding viruses that replicate in the cytoplasm, there is even a larger diversity of VF morphologies. Nucleo-cytoplasmic large DNA viruses (NCLDV) like Poxviridae, Asfarviridae and Iridoviridae, as well as dsRNA viruses like Reoviridae and ssRNA(-) viruses like Filoviridae, replicate in viroplasms, which are large electronic-dense inclusions located at the host cell cytoplasm [13]. Other viruses such as Picornaviridae and Nidovirales Arteriviridae and Coronaviridae induce the formation of double membrane vesicles (DMV), membranous structures of about 200–300 nm in diameter that are derived from the endoplasmic reticulum (ER) or Golgi membranes. Some other viruses replicate in membrane invaginations, 50–400 nm in diameter, called spherules. These spherules can appear on several enveloped cellular components, depending on the virus that induces them: ER for Flaviviridae and Alphaflexiviridae [14], mitochondria for Nodaviridae [15], chloroplasts for Tymoviruses [16], endosomes and lysosomes for Togaviridae [5] and peroxysomes for Tombusvirus [17]. Finally, other viruses induce the formation of VFs with rather specific morphologies, such as the tubular membrane structures that Bunyaviridae produce around the Golgi [18] or the giant volcano-shaped viral factories of the Mimivirus, which can grow to a size similar to that of the nucleus of the infected host cell [19].

Given their crucial role and their distinctive morphology, the observation of VF structures can be considered as a hallmark for proliferative viral infection and is in fact used as a diagnostic tool for certain viral infections [20]. Recently, VFs have gained interest as targets for antiviral therapies, as more factors involved in VF formation are being identified. Remarkably, the fact that the same cell compartments and signaling pathways are used by different viruses to generate their VFs paves the way for the development of broad-spectrum antiviral therapies [21].

Up-to-date cell-scale imaging techniques are thus fundamental to characterize these virus-induced alterations in the host cell substructures and, ultimately, to understand the multiple steps involved in the viral life cycle. In this context, cryo–soft X-ray tomography (cryo-SXT) stands out as a powerful tool for current virology research, providing 3D information on cellular infection processes and helping to understand drug action at the cellular level [22]. Cryo-SXT is so far the only available imaging technique that can yield nanometer-resolution 3D maps from vitrified whole-cell samples, thus avoiding chemical treatment or sectioning of the sample (and the potential artefacts that come with these treatments) [23,24]. Cryo-SXT provides complementary information to other biological imaging techniques, such as electron microscopy, X-ray fluorescence and visible light fluorescence, and is generally used as a partner method for 2D or 3D correlative imaging at cryogenic conditions in order to link function, location and morphology. Cryo-SXT is available at HZB X-Ray Microscopy Beamline U41-PGM1-XM at BESSY II (Berlin, Germany) [25], 2.1 beamline (XM-2) at the Advanced Light Source (Berkeley, CA, USA) [26], B24 beamline at Diamond Light Source (Oxford, UK) [27] and BL09-MISTRAL beamline at ALBA synchrotron (Barcelona, Spain) [23].

This review describes the fundamentals of cryo-SXT and the procedures for sample preparation. Several examples of cryo-SXT research on virus-infected cells are also shown.

## 2. Soft X-ray Tomography

### 2.1. Method Basics

Cryo-SXT is a synchrotron-based technique that uses soft X-rays to image samples in the water window energy absorption range (284 to 543 eV). At these energies, carbon-rich biological structures absorb X-rays heavily, especially compared to their oxygen-rich hydrated environment. Since attenuation of the photons travelling through the specimen depends on the thickness and the biochemical composition of the structures being imaged, the photon intensity values recorded for each voxel will reflect the chemical elements and their concentration in that voxel [28,29]. This is then used to discriminate the different organelles and subcellular structures in the obtained 3D data.

Data is collected as tomograms, each consisting of a set of transmission images collected at different tilt angles of the sample. Typically, acquisition of a tomogram takes about 5–30 min, depending on the exposure time.

Depending on the optics used, cryo-SXT can deliver a lateral resolution of up to 25 nm half pitch [22,30], thereby bridging the resolution gap between visible light diffraction-limited fluorescence light microscopy and electron microscopy. The achieved resolution is also dependent on the chosen data-collection scheme (i.e., if full or partial tilt series are performed) and the achieved signal-to-noise values. The choice of data-collection scheme depends on the sample holder: capillary tubes, such as those used in XM-2 beamline, can be rotated 360 degrees (full tilt), while microscopy grids used in the rest of beamlines can only be tilted ~140 degrees (partial tilt). On the other hand, signal-to-noise values will depend on the thickness of the biological samples, for which 10 μm is usually considered as the maximum compatible.

Cryo-SXT is frequently used in combination with other techniques, such as visible light fluorescence microscopy or transmission electron microscopy (TEM), following correlative approaches to guide location of specific features or events (such as viral particles or VFs) within the 3D cellular environment. For the same purpose, cryo-SXT microscopes are also equipped with an online visible light (fluorescence) microscope, which assists the user in choosing the areas of interest in the sample from which the cryo-SXT data is to be collected.

The following sections outline the procedures for sample preparation, data collection and data analysis for cryo-SXT experiments. For more detailed protocols, please refer to Groen et al. [24].

### 2.2. Sample Preparation for Cryo-SXT

The experimental workflow for cryo-SXT methodology starts with the seeding or deposition of cells on carbon-coated microscopy grids, previously sterilized by UV irradiation (Figure 1a). Alternatively, cells can be housed in thin-walled glass capillary tubes, which allow for the specimen to be rotated through 360 degrees [31]. Grids used for cryo-SXT have a mesh dividing the grid area into different regions, each of which is identified by an alphanumerical code that facilitates identification of the regions of interest during data collection. For some cell types with low cellular adhesion efficiency, plasma hydrophilization or poly-L-lysine functionalization of the grid surface may be required before seeding.

Seeded grids are then incubated in normal culture media until at least several cells have grown in each mesh square of the grids (usually 1–2 days after seeding) (Figure 1b). For virus research, cells are then inoculated with virus stock, and infection is allowed to proceed by further incubation of the grids at the required biosafety culture conditions.

At the designated time points after culture inoculation with the virus stock, infection is stopped by vitrification of the grids for virus classified as biosafety level 1 (BSL1). For viruses classified as biosafety level 2 (BSL2), such as HIV, dengue virus, Zika virus and rabies virus, which are not generally transmitted by airborne routes, the blotting and vitrification requires handling inside BSL2 laboratories and containment level 2. For viruses requiring containment level 3, typically those with a potential for respiratory transmission and/or those that can cause serious or fatal illness, such as SARS-CoV-2, HPAI H5N1, Japanese encephalitis and West Nile virus, higher containment measures are needed, including additional primary and secondary barriers to reduce or eliminate the release of infectious organisms into the immediate laboratory space and the general outside environment. For those viruses, the infected grids must be inactivated at the BSL3 facility before moving them to the beamline. This can be accomplished, for instance, by chemical fixation using 2% formaldehyde for 10 min [32].

Finally, non-infectious, inactivated samples are washed out, and grids holding the samples are cryopreserved. In conventional EM fixation, samples are prepared by chemical fixation with osmium tetroxide or aldehydes at room temperature. Then, the sample is dehydrated with organic solvents, infiltrated and embedded in epoxy resin. This process induces artefacts such as distortion of the cell morphology, which makes it inadequate for preserving membrane-rich structures and can lead to misinterpretations of cellular structures [33,34]. A much better alternative for preservation of cellular structures is cryofixation of the specimen, either by high-pressure freezing (HPF), which combines jets of liquid nitrogen with very high pressures, or ultra-rapid cryofixation by plunge freezing, which involves rapid immersion of the grid in a cryogen. Both techniques obtain vitrified biological samples without the formation of ice crystals, which could destroy the sample, or containing ice crystals small enough not to damage the ultrastructure of the cells [35]. During vitrification, the sample is frozen at a very high cooling rate, instantly fixing the sample in an amorphous, glass-like phase and thus keeping all cellular components in a close-to-life state.

For thin samples of less than 20 μm thickness, as is the case for cryo-SXT studies, plunge freezing vitrification is currently the method of choice. In this case, the excess of solution on the grid holding the sample is removed by blotting, right before the grid is rapidly plunged into liquid ethane. Plunge-freezing vitrification of cryo-SXT samples is relatively straightforward, especially in comparison to sample preparation for cryo-EM, since cryo-SXT is less sensitive to devitrification processes affecting areas smaller than ~10 × 10 nm^2^. In general, when plunge-freezing cryo-SXT samples, the most critical parameter that needs to be optimized is the blotting time.

To facilitate data analysis and alignment of the tomograms, gold particles 100–250 nm in diameter, used as fiducial markers, are usually added to the grids before blotting. Likewise, fluorescent stains targeting cellular organelles can be added to the cell cultures before washout of the culture media, to ease the identification of cells on the grids and the determination of the regions of interest. After vitrification, grids must be kept at cryogenic temperatures at all times.

Vitrified samples can then be screened by imaging in cryo conditions, using a visible light microscope equipped with a cryo-stage. In this step, the quality of the samples is assessed by checking the integrity of the carbon support film, the thickness of the ice and the homogeneous distribution of cells and fiducial markers on the grid. Likewise, a fluorescent signal can be used to locate isolated cells suitable for cryo-SXT imaging. If fluorescent markers (such as an mCherry tag) are fused to any of the viral proteins of the virus strain used for infection, infected cells can be identified by the presence of the fluorescent signal, and their location can be registered for posterior analysis with cryo-SXT.

### 2.3. Imaging at the Transmission X-ray Microscope

The selected grids are loaded into the transmission X-ray microscope (TXM) by means of a transfer station, which allows transfer from the storage cryobox to the microscope sample shuttle while maintaining the cryogenic conditions [23,36] (Figure 1c,d).

Once in the TXM (Figure 1e), the grids are first imaged using an on-line visible light microscope, in brightfield and/or fluorescence modes, to select the grid squares that will be further analyzed with X-rays (Figure 1f–h). Then, 2D X-ray mosaics of the selected grid squares are acquired (Figure 1i). These 2D mosaics are used to select the areas in which tomography will be performed. For each region of interest being imaged, a set of images is collected, each taken at a different tilt angle (Figure 1h). The typical angular range for these tomograms is ±70 or ±60 degrees, or up to 360 degrees for samples in capillary tubes.

Each projection of the image stack is normalized by a factor linked to the constant *c_ff_* (related to beamline optics geometry) and the electron current of the synchrotron storage ring [23,36], and then the different projections are aligned using the positions of the gold fiducials, reaching subpixel accuracy [37,38]. Several software applications can be used for this step, although the highest possible accuracy in the alignment is usually reached by manual alignment. Aligned and normalized stacks are then used to produce the reconstructed volumes [39], which can be further analyzed and segmented with specific software tools [37,40,41,42] (Figure 1i,j).

## 3. Examples of Virus Research with Cryo-SXT

Several virus-related studies have been performed with different soft X-ray microscopes worldwide (e.g., [43,44,45,46,47,48]). In this section, three such applications performed at the MISTRAL beamline are described, to illustrate the potential of cryo-SXT for virology research.

### 3.1. Cryo-SXT Analysis of Ultrastructural Alterations in Liver Cells upon HCV Infection and after Antiviral Treatment

Hepatitis C virus (HCV), a member of the Flaviviridae family, has an enveloped, spherical capsid ~50 nm in diameter that protects a linear ssRNA(+) genome. HCV infection is the cause of severe liver disease in millions of humans worldwide and remains the leading cause for liver transplantation worldwide [49]. Recently, interference of HCV replication with cellular homeostasis has been raised as one of the drivers that influence HCV pathogenesis and progression of the disease [50].

To investigate the ultrastructural alterations induced by HCV replication, cryo-SXT was performed on a model of HCV infection, yielding native, whole-cell, three-dimensional (3D) maps of the infected cells. For this, Huh-7 cells constitutively expressing an HCV-dependent fluorescent reporter, which can be used as a marker for viral replication [51], were inoculated with spread-deficient infectious HCV virions (HCVtcp) [52], capable only of single-cycle HCV infections, used as a surrogate model of HCV infection to comply with biosafety regulations. The infected cultures were vitrified at early (16–24 hpi) and late (72 hpi) infection stages. These samples, together with control (uninfected) cells, were subjected to cryo-SXT at BL09-MISTRAL beamline (ALBA). The reconstructed volumes were used to identify and segment different subcellular compartments (Figure 2a–f) [22].

In all the studied samples, the cell nuclei appeared as granular structures corresponding to chromatin fibers, and nucleoli could also be identified; in fact, no significant structural differences were found in the nuclei of the different samples, indicating that HCV replication does not trigger any visible structural changes in the nucleus. Regarding cytoplasmic structures, however, the tomograms showed larger differences. In control cells, the ER appeared as a tubular structure with continuous cisternae, surrounding mitochondria that displayed a normal morphology (Figure 2a). In contrast, for cryo-SXT volumes of cells in early infection stages, HCV infection induced the formation of heterogeneous vesicle-like structures in an intricate network of vesicular and tubular structures, likely derived from the ER (Figure 2c). These particular membranous compartments have been shown to host viral replicase components and are likely the site for HCV RNA replication [53]. The membranous web compartment in HCV replicating cells constitutes the most prominent virus-induced alteration in the cellular architecture [54]. Another important differential feature of the HCV modification is the aberrant mitochondrial morphology: abnormal mitochondria were also identified in the tomograms, characterized by an increased absorption contrast that suggests matrix condensation, as well as alterations in the number of visible cristae compared to normal mitochondria. The vesicles and tubes of the membranous web were localized in discrete cytoplasmic areas, surrounded by areas that displayed an apparently normal ultrastructure. It is noteworthy that most of the mitochondria that surrounded the modified ER areas displayed morphological modifications (Figure 2c). In particular, analysis of cryo-SXT tomograms of infected cells revealed that 72.7% of mitochondria displayed aberrant morphologies (compared to 7.6% in control cell cultures) [22].

In cells at a late stage of HCVtcp infection, the cytoplasm was completely modified: a mature membranous web contained dilated tubules and vesicles and long tubular structures of heterogeneous size and different spatial orientation, conferring a sponge-like appearance to the entire cytoplasmic volume (Figure 2e).

In addition, to determine if such HCV-induced ultrastructural alterations could be reverted by antiviral drugs, cells bearing a subgenomic HCV replicon were treated for 7 days with a combination of clinically relevant antivirals, sofosbuvir + daclatasvir [55], before being vitrified and imaged at BL09-MISTRAL beamline. Cryo-SXT of treated samples revealed a drastic reduction in the volume of the membranous web compared to non-treated cells, with just 4% of abnormal mitochondria being still noticeable in the whole-cell cytoplasm [22] (Figure 2g). The study by Perez-Berna et al. showed that such remaining ultrastructural alterations could still be observed in treated samples, whereas viral protein expression was undetectable by Western Blot in equivalently treated samples [22], proving that cryo-SXT can be more sensitive than biochemical assays for characterization of ultrastructural alterations in infected cells. This study also highlighted the potential of cryo-SXT in assisting drug development at a preclinical level, by determining the potential impact of candidate compounds on the ultrastructure of the cell.

### 3.2. X-ray Tomography Analysis of Vaccinia Virus Morphogenesis

Poxviruses are large, enveloped, DNA viruses that replicate entirely in the cytoplasm [56]. The Poxviridae family comprises viruses that infect vertebrate and invertebrate species, including variola virus, the causal agent of smallpox disease in humans. The type species of this viral family, vaccinia virus (VAVC), has a relatively large particle (around 250 nm in diameter and 350 nm in length) with a characteristic brick shape, which results from a very complex morphogenesis. The process starts with the formation of large, dense bodies of viroplasm (VP) within the VFs, which are located in the host cell cytoplasm. These VPs, which contain the virus core proteins and the viral DNA, incorporate sheets of membrane scavenged from the ER. Assembly of a viral protein scaffolds on these membrane segments forces their curvature and extension and leads to the formation of crescents, which further enlarge to form spherical immature virions (IV). IVs then follow a complex maturation process, involving reorganization of viral envelopes, formation of a distinct core and a drastic change in the overall structure, ultimately yielding the infectious form, the brick-shaped intracellular mature virus (MV). A subset of these MVs further proceeds in the maturation process by interacting with the trans-Golgi network to acquire two additional membrane bilayers, giving rise to the intracellular enveloped virus (IEV). These can exit the host cell by exocytosis, losing the outermost membrane bilayer, yielding the extracellular enveloped virus (EEV) (reviewed by Moss B. [57]).

Such a complex maturation process, involving multiple sets of temporal interactions between cellular membranes and organelles with viral components, takes place in extended regions of the cytoplasm of the infected cell. This makes it difficult to study with high-resolution imaging techniques, mainly because the information is retrieved from a small percentage of the volume of the whole cell (since, for instance, cryo-ET is restricted to samples that are well below 800 nm in thickness, from a total cell thickness of 10–20 μm), thus preventing a quantitative analysis of the viral life cycle at the cellular level.

To circumvent this problem, VACV-infected cells were studied by cryo-SXT. For this, PtK2 cells were infected with a GFP-expressing VACV strain and vitrified by plunge freezing. The infected cells were selected by visible light fluorescence microscopy of the GFP marker and subsequently imaged using the X-ray microscope under cryogenic conditions at beamlines U41-PGM1-XM (BESSY II) and BL09-MISTRAL (ALBA). Tomographic tilt series of X-ray images were used to generate three-dimensional reconstructions, with an estimated 3D-resolution of up to 55 nm [58]. This resolution was sufficient to detect two types of virus particles, differing in shape, size and density, which were identified as VACV IV and MV (Figure 3), as confirmed by TEM on the same samples.

Remarkably, since cryo-SXT can yield the reconstruction of whole cell volumes, a compartmentalization of the viral particles within the VFs according to their maturation stage could be observed: IVs aggregate in areas near the nuclei, while MVs are grouped towards the periphery of the cell (Figure 3c). Closer examination of the VFs and the viral-related particles in their neighborhood revealed the presence of VP coupled to crescent-like and incomplete IVs (Figure 3d, left panel), as well as transition zones where both types of particles were found (Figure 3d, middle-left panel) and MVs accumulations near the plasma membrane of the cell (Figure 3d, right and middle-right panels).

### 3.3. Ultrastructure Alterations Associated with Zika Virus Replication

Zika virus (ZIKV) is a mosquito-borne virus that gained the attention of the scientific and healthcare communities due to the epidemic that swept through South and Central America in 2014–2016, owing to the association of this virus with the occurrence of congenital deformities, particularly microcephaly in infants born to the infected mothers, and other severe neuropathies such as Guillain-Barré syndrome in children and adults [59,60].

ZIKV belongs to the Flaviviridae family and is closely related to the other Flaviviruses, including dengue virus, West Nile virus, yellow fever virus and chikungunya virus. The virus has an enveloped, icosahedral capsid about 50 nm in diameter, harboring a monopartite, linear (+)ssRNA genome.

A previous study combining immunofluorescence microscopy with electron tomography showed ZIKV infection in human hepatoma and neuronal progenitor cells induces a drastic structural modification of the cellular architecture, leading to the formation of viral replication factories surrounded by a network of microtubules and intermediate filaments [61]. As described for other Flaviviruses, the virus replication activity is sheltered in replication factories: vesicles formed by invaginations of ER membranes, clustered in the so-called vesicle packets [53].

To obtain further details of the cellular context of these ultrastructure alterations, human glioblastoma U251 cells were grown on support grids and infected with Zika virus American strain. After vitrification by plunge freezing, the cells were imaged at BL09-MISTRAL beamline (ALBA). Most of the analyzed cells displayed signs of infection, including a rounded cell shape, a kidney-shaped nucleus and a largely disrupted ER (Figure 4).

Accordingly, analysis of the acquired tomograms allowed the identification of vesicles ~150 nm in diameter in the lumen of the distorted ER, which could be recognized as ZIKV replication vesicles (Figure 4c–f). Some of these vesicles contained one discrete heavily-dense particle, compatible in size with a ZIKV virion (~50 nm). Other structural alterations of the ER were observed, forming convoluted membranes and, more rarely, membrane tubules. Comparison of the tomograms acquired from ZIKV-infected cells with those obtained from mock-infected control cells highlighted virus-induced changes in the mitochondria disposition, with these no longer being dispersed throughout the cytoplasm in the virus-infected samples but rather concentrated around the replication vesicles (Figure 4b). In addition, large perinuclear granules containing amorphous dense material were also observed in the infected cells, similar to what has been reported for DENV infections [62].

## 4. Discussion

In virus-related research, structural cell biology methods allow visualization and understanding of the reorganization of cellular structures and the formation of new virus-induced ones during virus infection processes. This has traditionally been studied by visible-light microscopy or by electron microscopy or, more often, by a combination of both. Visible-light microscopy is used to study in vivo samples, to determine the location of individual molecules inside a cell or to identify and quantifying organelles, but sub-cellular nanometric resolution (achievable with super-resolution microscopy methods) is limited to imaging of fluorescently labelled molecules and does not include the surrounding cell environment [63]. On the other hand, transmission electron microscopy can yield very high spatial resolution images (in the order of few nanometers) of cellular structures [64], but the low penetration power of electrons in biological materials limits the thickness of the specimen to be studied to 500–800 nm [65].

In this sense, the relatively recent application of soft X-ray transmission microscopy to mesoscale imaging of biological samples can be seen as the bridge that covers the gap between the low-resolution (and high-penetration) visible light imaging techniques that provide dynamic information of specific processes, and the high-resolution (but low-penetration) electron microscopy ones that achieve molecular resolution [66]. Cryo-SXT benefits from two intrinsic properties of soft X-rays in the water window energy range (284−543 eV), that is, between the carbon and the oxygen K absorption edges: (i) a high natural contrast between carbon-based structures and the water in their surroundings and (ii) a high penetration depth in biological (hydrated) samples, up to ~10 µm.

The examples in Section 3 of this review show how cryo-SXT can be used to obtain the 3D map of a viral factory structure, imaged from whole, vitrified, infected cells. These tridimensional maps have sufficient resolution to distinguish details of the morphological features of the cell substructures. As the whole cell can be imaged, quantitative data related to these morphological details can be generated, upon analysis of the reconstructed cell volume. This is illustrated by the research described in Section 3.1, in which cryo-SXT was used to assess the effectivity of different antiviral drugs in reverting the mitochondria ultrastructural aberrations induced by HCV infection. In this study, the morphology of >500 mitochondria (obtained from ~10 whole-cell volumes) was characterized for each sample type (control cells, non-treated and treated) [22]. Other examples of quantitative cryo-SXT studies include the measurement of the average volumes of the nuclei at different times of HSV-1 infection [45], the determination of the number of mitochondria in SARS-CoV-2–infected cells [43] and the statistical characterization of the 3D configuration of centrioles in human primary CD4 T cells [67].

Remarkably, such quantitative studies are achievable thanks to the relatively high throughput of the cryo-SXT pipeline: considering a typical acquisition time of 5–30 min per tomogram, and that 5–10 tomograms are usually required to cover the whole cell area, data collection for a set of 10 whole-cell volumes represents about one full day of beamtime. In addition, sample preparation for cryo-SXT only requires 2–3 h per sample type (not counting cell growth and infection times), thus representing a less significant burden, especially when compared to other more time-demanding sample preparation pipelines for methodologies such as focused ion beam tomography (FIB-SEM). FIB-SEM is a powerful technique for 3D imaging of sub-micron features in a sample, in which the specimen is sequentially milled away using an ion beam while the newly exposed surface is imaged using an electron beam, yielding reconstructed volumes with resolutions <10 nm in all directions [68]. However, this process is slow and requires the sample to be fixed in a block, which can lead to significant sample distortions [34]. To bypass the need for chemical fixation, cryo-FIB-SEM has recently been developed to image vitrified biological samples at cryogenic temperatures. Nevertheless, further studies are still required before cryo-FIB-SEM can be established as a standard method for whole-cell imaging [69].

The resolution of the obtained cryo-SXT tomograms mainly depends on the zone plate (ZP) installed in the beamline optics. The ZP basically acts as an “objective lens”, gathering the photon beam transmitted through the sample and focusing the magnified image of the specimen onto the detector. The resolving power is proportional to the smallest outermost zone width of the zone plate, which defines the higher achievable resolution limit. Each cryo-SXT beamline has chosen the specifications of their ZP as a compromise between resolution, field of view and depth of field; the smaller the width of the ZP outermost zone, the higher the achieved resolution, but also the smaller the associated field of view and depth of field. U41-PGM1-XM beamline (BESSY II) is equipped with a 25 nm outermost zone width ZPs [25]; both B24 beamline (DLS synchrotron) and MISTRAL beamline (ALBA synchrotron) use 25 nm and a 40 nm outermost zone width ZPs [47,70,71]; and XM-2 beamline (Advanced Light Source) is equipped with 50 nm and 80 nm outermost zone width ZPs [26].

In some cases, for viruses larger than ~50 nm in diameter, the viral particles themselves can also be observed. Given the sizes of the (currently known) viral particles, ranging from the 17 nm in diameter of the PCV capsid [72] to lengths of up to ~1.9 µm of Ebolavirus filaments [73] or ~1.5 µm of the gigantic pithovirus [74], the 50 nm limit should still allow for the visualization of virions for most viral families. As shown in Figure 3d, for the largest and most complex viruses, such as poxviruses, cryo-SXT yields data at sufficient resolution to distinguish between different virion maturation stages.

Every combination of virus and cell type is unique and can therefore show differences in ultracellular structure, morphology or absorption. Cryo-SXT can provide understanding of the spatial organization of the different cellular organelles during the infection, but to achieve a better comprehension of the complexity of virus–host interactions, it is recommended to combine cryo-SXT data with other imaging techniques that can provide the precise localization or the higher resolution structure of the particular viral or host factors being studied. Correlative workflows combining cryo-SXT with techniques such as cryo-ET, visible light fluorescence microscopy, hard X-ray fluorescence tomography or numerical simulations have already proven their potential in several multi-length scale structural studies [24,43,44,45,46,75,76,77]. A particularly powerful approach is a correlative cryo-imaging pipeline, in which the same sample is imaged with super-resolution 3D structured illumination fluorescence microscopy (cryo-SIM) and with cryo-SXT, allowing for a comprehensive view of both cellular ultrastructure and the related molecular organization over extended cellular volumes [78,79]. This pipeline has been recently established at Diamond Light Source B24 beamline and will also be available in the near future at ALBA synchrotron MISTRAL beamline. An example of the contributions of such a correlative cryo-SIM/cryo-SXT approach is its application to the characterization of the reovirus release pathway from intracellular vesicles during the early stages of infection [47].

In summary, cryo-SXT is a powerful imaging tool for the assessment of complex cellular processes in whole, intact, hydrated cells that can be exploited for the study of viral infections and to aid in the development of novel antiviral therapeutics.

## Figures and Tables

**Figure 1 viruses-13-02109-f001:**
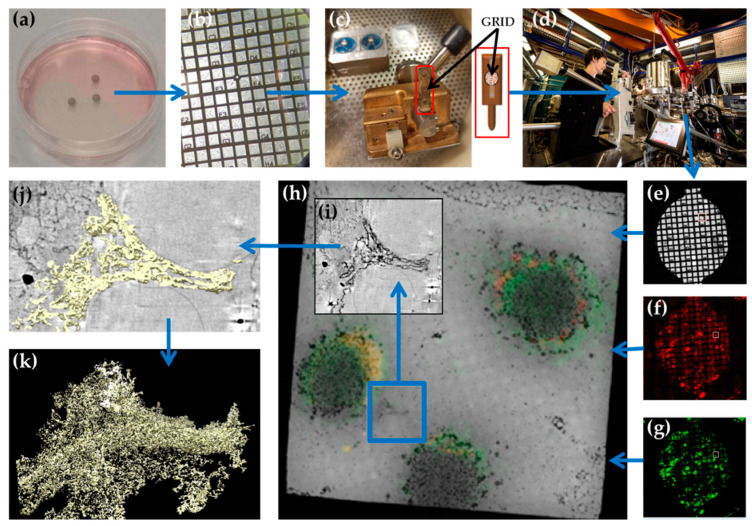
Workflow followed in cryo-SXT. (**a**) P60 Petri dish with 3 grids on which cells were seeded. (**b**) Close-up view of cells grown on top of a grid, 24 h after seeding. (**c**) Workstation with the grid sample holder inserted into the loader. In the picture, the grid is already loaded into the grid sample holder (red box). A schematic drawing of a sample holder is shown on the right. The position of the loaded grid is also indicated. (**d**) Attachment of the transfer chamber to the transmission X-ray microscope (TXM). (**e**–**g**) Mosaics of the grid acquired with visible light (**e**), red fluorescence (**f**), and green fluorescence (**g**). (**h**) 2D X-ray mosaic of a grid square, superposed with the correlated red and green fluorescence mosaics, showing the positions of 3 cells. (**i**) Slice from the reconstructed 3D volume corresponding to the area marked with the white box in (**h**). (**j**) Manual segmentation of the surface boundaries identifying a polysome organelle present in the cell. (**k**) Manual segmentation of the organelle in (**j**). Panels (**a**) to (**d**) adapted from Groen et al. 2021 [24], with permission from JoVE).

**Figure 2 viruses-13-02109-f002:**
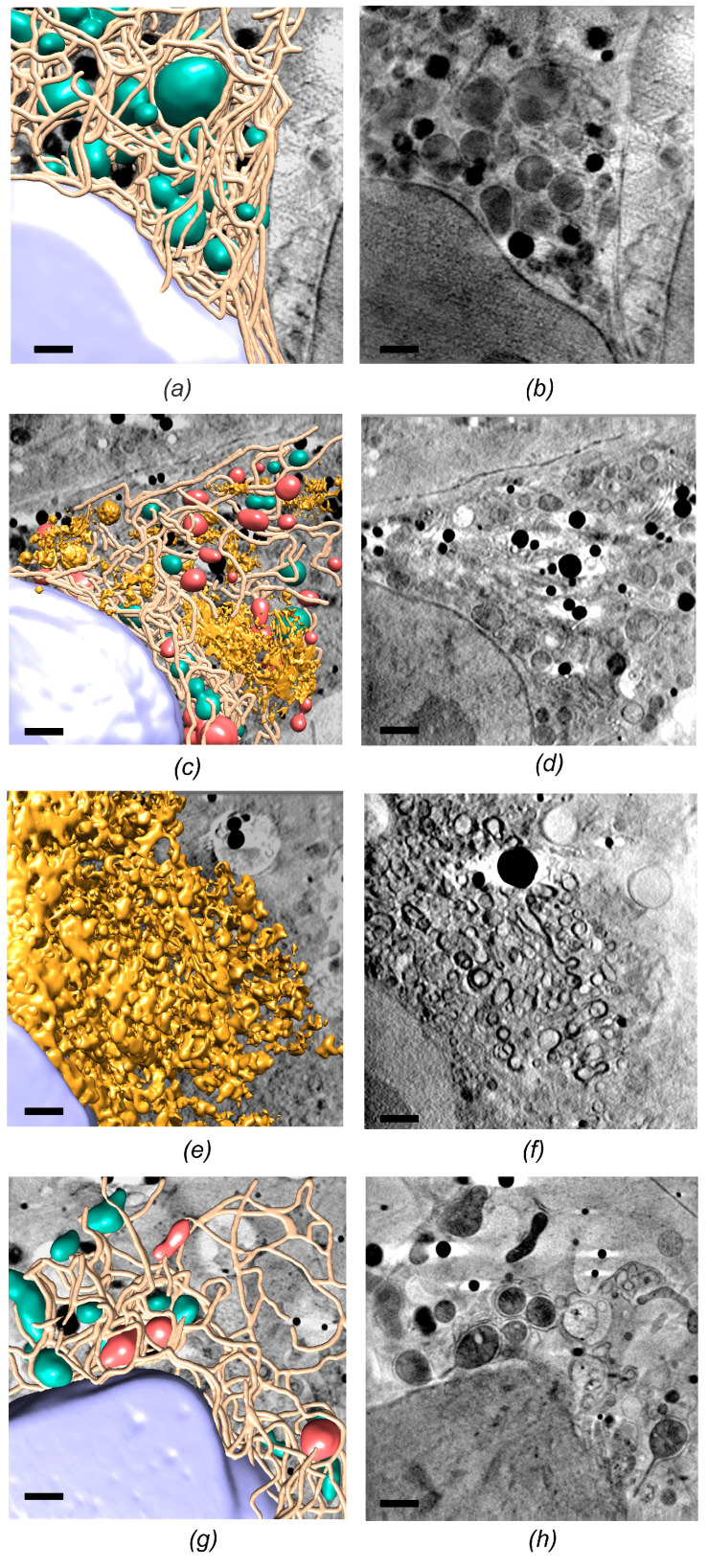
Three-dimensional reconstruction of whole-cell volumes by cryo-SXT. Control (**a**,**b**); HCVtcp-infected cell cytoplasm at different stages of HCV infection: early infection (**d**,**e**) and late infection (**e**,**f**); and HCV replicon-bearing cell lines treated with antiviral compounds for 7 days (**g**,**h**). Manual segmentation of the surface boundaries identifying organelles is shown in the left column (**a**,**c**,**e**,**g**), with normal mitochondria displayed in green, abnormal mitochondria in red, nuclear envelope in purple, endoplasmic reticulum in brown, and membranous web in gold. Volume slices of the tomograms are shown in the right column (**b**,**d**,**f**,**h**). Scale bar corresponds to 1 μm.

**Figure 3 viruses-13-02109-f003:**
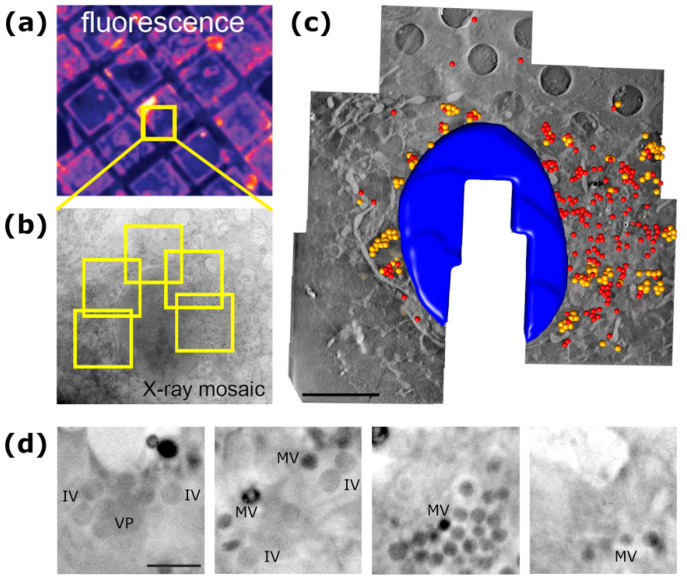
Cartography of the vaccinia virus infection by cryo-SXT. (**a**) Infected cells were located by visible light fluorescence cryo-microscopy. (**b**) X-ray mosaic shows the whole-cell 2D projection used to configure the area of interest. Several tomograms were acquired from the same cell (yellow squares) to obtain a whole-cell tomogram. (**c**) Segmented 3D volume of the whole-cell tomogram. Nucleus is shown in blue; immature virions and mature virions are shown as yellow and red spheres, respectively. Scale bar corresponds to 5 µm. (**d**) Close-up views of different tomograms, where viroplasm (VP), immature virions (IV) and intracellular mature virions (MV) can be observed. Scale bar corresponds to 0.5 µm. Adapted from Chichón et al. [58], with permission from Elsevier.

**Figure 4 viruses-13-02109-f004:**
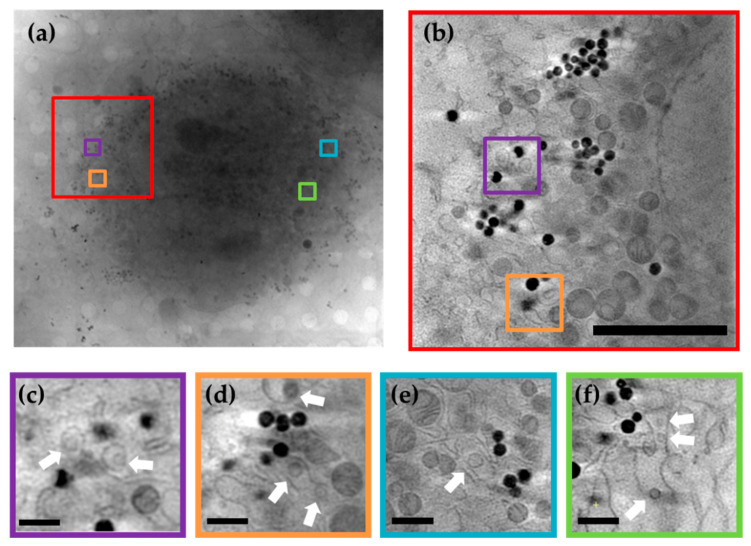
Imaging of Zika virus-induced ultracellular alterations by cryo-SXT. (**a**) X-ray mosaic showing a general view of two Zika virus-infected cells. (**b**) Frame corresponding to the 0º rotation of the tomogram acquired at the region indicated in red in (**a**). Scale bar corresponds to 5 µm. (**c**–**f**) Close-up views of different tomograms acquired from the cells in (**a**), in which a convoluted ER membrane and packets of replication vesicles (white arrows) can be observed, together with the associated mitochondria and electron-dense granules. Scale bar corresponds to 0.5 µm.

## Data Availability

Not applicable.

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
