# Peer review of "Imaging of Virus-Infected Cells with Soft X-ray Tomography"

_viruses, 2021, doi:10.3390/v13112109_

Round 1

Reviewer 1 Report

The authors have described how viruses can be imaged using soft X-ray tomography. They provide an overview of the underlying principles and general procedure for imaging by soft X-ray tomography, and then they illustrate the application of the method to imaging of various viruses. This is a nice introduction to the technique, and it illustrates how the approach can be useful in the study of viruses.

I like this review and I think it can be published with only minor changes.

  1. There were a number of small English mistakes. I have corrected them, and attach a Word document with the changes.
  2. I would suggest dropping "at the MISTRAL beamline" from the title. This makes the article sound very specific.  I think it would be good somewhere else in the text (maybe right at the beginning of section 3) to cite some virus work from other beamlines, such that readers can see that the work is in general possible at other soft X-ray microscopes, but there is no need to go into detail about those other studies. I just imagine a few sentences something like: Virus analysis has been performed at several different soft X-ray microscopes worldwide (citations here). In this article, we illustrate the approach with several applications at the MISTRAL beamline.
  3. Line 185 - "The image stacks are normalized and corrected by the machine current". I doubt that many readers will know what is meant by the machine current, and I myself am confused about exactly what is done. Does it mean that each tilt-angle image is normalized by the measured illumination intensity of the light source? In any event, I think the sentence could be made simpler and clearer.

Author Response

Response to comments from Reviewer 1:

  1. There were a number of small English mistakes. I have corrected them, and attach a Word document with the changes.

We thank the reviewer for spotting the mistakes and for taking the time to correct them; we have included most of the suggested changes in the updated version.

  1. I would suggest dropping "at the MISTRAL beamline" from the title. This makes the article sound very specific. I think it would be good somewhere else in the text (maybe right at the beginning of section 3) to cite some virus work from other beamlines, such that readers can see that the work is in general possible at other soft X-ray microscopes, but there is no need to go into detail about those other studies. I just imagine a few sentences something like: Virus analysis has been performed at several different soft X-ray microscopes worldwide (citations here). In this article, we illustrate the approach with several applications at the MISTRAL beamline.

This is a shared view from both reviewers, so we have followed the suggestion and included work from other cryo-SXT beamlines in the text.

In particular, as suggested by the reviewer, a new sentence has been included at the beginning of section 3:

“Several virus-related studies have been performed at different soft X-ray microscopes worldwide (e.g. [36–40]). In this section, three of such applications performed at the MISTRAL beamline are described, to illustrate the potential of cryo-SXT for virology research.”

In addition, some of these studies have been further discussed in section 4:

“Other examples of quantitative cryo-SXT studies include the measurement of the average volumes of the nuclei at different times of HSV-1 infection ([38]), or the determination of the number of mitochondria in SARS-CoV-2-infected cells ([36]).“

“An example of the contributions of such correlative cryo-SIM/cryo-SXT approach is its application to the characterization of the reovirus release pathway from intracellular vesicles during the early stages of infection [40].”

Accordingly, we have also modified the title to remove the reference to MISTRAL beamline. The title now reads: “Imaging of Virus-Infected Cells with Soft X-ray Tomography”.

  1. Line 185 - "The image stacks are normalized and corrected by the machine current". I doubt that many readers will know what is meant by the machine current, and I myself am confused about exactly what is done. Does it mean that each tilt-angle image is normalized by the measured illumination intensity of the light source? In any event, I think the sentence could be made simpler and clearer.

This sentence was indeed not clear, and we thank the reviewer for pointing it. The sentence has been rewritten and now reads:

“Each projection of the image stack is normalized and corrected by a factor linked to the constant cff (related to beamline optics geometry) and the electron current of the synchrotron storage ring [23,29]”

Citations in this sentence correspond to two articles describing the normalisation procedures with further detail:

Sorrentino, A.; Nicolas, J.; Valcarcel, R.; Chichon, F. J.; Rosanes, M.; Avila, J.; Tkachuk, A.; Irwin, J.; Ferrer, S.; Pereiro, E., MISTRAL: a transmission soft X-ray microscopy beamline for cryo nano-tomography of biological samples and magnetic domains imaging. Journal of synchrotron radiation 2015, 22

Pereiro, E.; Nicolas, J.; Ferrer, S.; Howells, M. R., A soft X-ray beamline for transmission X-ray microscopy at ALBA. Journal of synchrotron radiation 2009, 16 (Pt 4), 505-12.

Reviewer 2 Report

This review describes imaging virus-infected cells with soft x-ray tomography. Unfortunately, this article by Garriga et al fails on many levels. Firstly, it reviews two papers published by the group at the Barcelona synchrotron, ignoring publications of virus-infected cells imaged at other synchrotrons. For example:

  1. Mendonca et al. bioRxiv. 2020. doi: https://doi.org/10.1101/2020.11.05.370239. From the Diamond Light Source
  2. Kounatidis et al. Cell, 2020 Jul 23;182(2):515-530.e17. doi: 10.1016/j.cell.2020.05.051. From the Diamond Light Source; 
  3. Aho et al. Viruses. 2019, Oct; 11(10): 935. doi: 10.3390/v11100935. From the Advanced Light Source.
  4. Aho et al., Nature Sci. Rep. 2017, Jun 16;7(1):3692. doi: 10.1038/s41598-017-03630-y. From the Advanced Light Source.
  5. Myllys et al. Nature Sci. Rep. 2016, Jun 28;6:28844. doi: 10.1038/srep28844. From the Advanced Light Source.

Secondly, the authors present new data that has not been peer-reviewed and make unfounded claims. They treated cells with antiviral drugs for 7 days, and claim there is a “drastic reduction of the volume of membranous web compared to non-treated cells (no supporting quantitation of volume provided), although abnormal mitochondria are still noticeable in the cytoplasm.” They then conclude that “given that in equivalently treated samples, viral protein expression was undetectable by Western blot, cryo-SXT proved to be more sensitive than the biochemistry assays for characterization of ultrastructural alterations in infected cells.  There is no basis for this claim; they only provided x-ray images - no biochemistry or any other data were provided (this work has not been peer reviewed).

Thirdly, the authors claim that cryoSXT is superior to cryoET because the latter is restricted to samples well below 800 nm in thickness - a very small portion of the cell - preventing a quantitative analysis of the viral life cycle at the cellular level. However, this paper does not shown a single image of a whole cell nor any quantitative data at the cellular level. This suggests the techniques are comparable.

Finally, it is difficult to see that the work shown here provides any valuable new information. The specimen preparation involves all the tedium required for CryoET, the data are lower resolution, and the authors did not image whole cells to provide accurate quantitative information.

Author Response

Response to comments from Reviewer 2:

This review describes imaging virus-infected cells with soft x-ray tomography. Unfortunately, this article by Garriga et al fails on many levels. Firstly, it reviews two papers published by the group at the Barcelona synchrotron, ignoring publications of virus-infected cells imaged at other synchrotrons. For example:

  1. Mendonca et al. bioRxiv. 2020. doi: https://doi.org/10.1101/2020.11.05.370239. From the Diamond Light Source
  2. Kounatidis et al. Cell, 2020 Jul 23;182(2):515-530.e17. doi: 10.1016/j.cell.2020.05.051. From the Diamond Light Source; 
  3. Aho et al. Viruses. 2019, Oct; 11(10): 935. doi: 10.3390/v11100935. From the Advanced Light Source.
  4. Aho et al., Nature Sci. Rep. 2017, Jun 16;7(1):3692. doi: 10.1038/s41598-017-03630-y. From the Advanced Light Source.
  5. Myllys et al. Nature Sci. Rep. 2016, Jun 28;6:28844. doi: 10.1038/srep28844. From the Advanced Light Source.

As stated in the response to point #2 from Reviewer 1, we have included work in virus research from other cryo-SXT beamlines in the new version of the text.

We thank the reviewer for the suggested references; citations to these articles have been included in the updated text at the beginning of section 3, and some of them have been further described in section 4. (See answer to Reviewer 1, point #2 for details of the updated text)

Secondly, the authors present new data that has not been peer-reviewed and make unfounded claims. They treated cells with antiviral drugs for 7 days, and claim there is a “drastic reduction of the volume of membranous web compared to non-treated cells (no supporting quantitation of volume provided), although abnormal mitochondria are still noticeable in the cytoplasm.” They then conclude that “given that in equivalently treated samples, viral protein expression was undetectable by Western blot, cryo-SXT proved to be more sensitive than the biochemistry assays for characterization of ultrastructural alterations in infected cells.”  There is no basis for this claim; they only provided x-ray images - no biochemistry or any other data were provided (this work has not been peer reviewed).

The data on HCV-infected and sofosbuvir + daclatasvir-treated cells presented in section 3.1 of this review is taken from a previously published work by some of the co-authors of this manuscript (Pérez-Berna et al., Structural Changes In Cells Imaged by Soft X-ray Cryo-Tomography During Hepatitis C Virus Infection. ACS nano 2016, 10 (7): 6597-611). The referenced article, which was indeed been peer-reviewed before its publication, includes quantitative data and biochemistry analyses on the studied samples. As the reviewer pointed out, these details were omitted in the previous version of the manuscript.

The text in section 3.1 has been updated to clarify the origin of the data. The new text now reads:

“The study by Perez-Berna et al. showed that such remaining ultrastructural alterations could still be observed in treated samples, whereas viral protein expression was undetectable by Western Blot in equivalently treated samples (see Figure 8 in Perez-Berna et al. [22]), proving that cryo-SXT can be more sensitive than the biochemical assays for characterization of ultrastructural alterations in infected cells [22]”

Thirdly, the authors claim that cryoSXT is superior to cryoET because the latter is restricted to samples well below 800 nm in thickness - a very small portion of the cell - preventing a quantitative analysis of the viral life cycle at the cellular level. However, this paper does not shown a single image of a whole cell nor any quantitative data at the cellular level. This suggests the techniques are comparable.

Sections 3.1 and 3.2 show segmentation models obtained from whole cell tomographic reconstructions, although the reviewer is correct in that whole-cell images as such are not shown in the paper.

Examples of quantitative data obtained at cellular level have been included in the updated version of the text, in sections 3.1:

“In particular, analysis of cryo-SXT tomograms of infected cells revealed that 72.7% of mitochondria displayed aberrant morphologies (compared to a 7.6% in control cell cultures) [22].”

 “Cryo-SXT of treated samples revealed a drastic reduction in the volume of the membranous web compared to non-treated cells, with just 4% of abnormal mitochondria being still noticeable in the whole cell cytoplasm [22]”

And in section 4:

“The examples in section 3 of this review show how cryo-SXT can be used to obtain the 3D map of a viral factory structure, imaged from whole, vitrified, infected cells. These tridimensional maps have sufficient resolution to distinguish details of the morphological features of the cell substructures. And, as the whole cell can be imaged, quantitative data related to these morphological details can be generated, upon analysis of the reconstructed cell volume. This is illustrated by the research described in section 3.1, in which cryo-SXT was used to assess the effectivity of different antiviral drugs in reverting the mitochondria ultrastructural aberrations induced by HCV infection. In this study, the morphology of >500 mitochondria (obtained from ~10 whole-cell volumes) was characterized for each sample type (control cells, non-treated, treated) [22]. Other examples of quantitative cryo-SXT studies include the measurement of the average volumes of the nuclei at different times of HSV-1 infection [38], or the determination of the number of mitochondria in SARS-CoV-2-infected cells [36].”

Finally, it is difficult to see that the work shown here provides any valuable new information. The specimen preparation involves all the tedium required for CryoET, the data are lower resolution, and the authors did not image whole cells to provide accurate quantitative information.

As proven by this comment, our manuscript did not achieve to showcase the advantages and potential usages of cryo-SXT. We have rewritten the Discussion to address this, hopefully improving the message of the manuscript. The updated text now contains details on the cryo-SXT pipeline throughput and more examples on the results that can be achieved with this technique (i.e. quantitative studies, combination with other techniques and correlative imaging with cryo-SIM).

In addition to the new paragraph in section 4 reproduced in the answer to the previous comment, the other new parts of the text read:

“Remarkably, such quantitative studies are achievable thanks to the relatively high throughput of the cryo-SXT pipeline: considering a typical acquisition time of 5-30 minutes per tomogram and that 5-10 tomograms are usually required to cover the whole cell area, data collection for a set of 10 whole-cell volumes represents about 1 full day of beamtime. In addition, sample preparation for cryo-SXT only requires 2-3 hours of sample preparation per sample type (not counting cell growth and infection times), thus representing a less significant burden, especially when compared to studies involving powerful but more time-demanding sample preparation pipelines such as FIB/SEM.”

[…]

“Correlative workflows combining cryo-SXT with techniques such as super-resolution 3D structured illumination cryogenic fluorescence microscopy (cryo-SIM), or cryo-electron tomography (cryo-ET), visible light fluorescence microscopy, hard X-ray fluorescence tomography or numerical simulations have already proven their potential in other several multi-length scale structural studies [24,36–39,62–64]. A partic-ularly powerful approach is a correlative cryo-imaging pipeline in which the same sample is imaged with super-resolution 3D structured illumination fluorescence microscopy (cryo-SIM) and with cryo-SXT, allowing for a comprehensive view of both cellular ultrastructure and the related molecular organization, over extended cellular volumes [65,66]. This pipeline has been recently established at Diamond Light Source B24 beamline, and will also be available in the near future at ALBA synchrotron MISTRAL beamline. An example of the contributions of such correlative cryo-SIM/cryo-SXT ap-proach is its application to the characterization of the reovirus release pathway from in-tracellular vesicles during the early stages of infection [40]. “

Reviewer 3 Report

Overall the quality of the review is good. the authors present a nice overview of a under rated technique but could benefit from a some relatively minor  modifications. 

Page 2 line 88: one more beamline offers SXT, the XM-2 beamline at Lawrence Berkeley National Laboratory

Page 3 line 111: it would be nice to explain why one would choose partial versus full tilt and which beamlines around the world offers what type of tilt

Page 3 line 127: Samples can also be housed in capillary tubes for full tilt rotation since the EM grids support doesn't allow for full tilt. 

Page 4 line 151: a short discussion of the potential effect of fixation on samples that are supposed to be in their native state (cryofixation) would be nice as well as a comparison in this case of sample fixed for block face imaging which would lead to similar resolution. 

Page 4 line 152: reference 28 seems inadequate, this reviewer cannot find any reference to 2 % formaldehyde in this citation.

Page 4 line 160: adding the vitrification process would improve the manuscript since the vitrification is the stepping stone for sample quality. 

Page 4 figure 1: this figure seems to be a little confusing, at least for panel c to e with reduced image size where no real details can be seen. a schematic representation of the workflow would work better. 

Page 5 line 194: the LBNL beamline has  a 360 degres stage

Page 6 line 247: "figfigffure" should be replaced by "fig"

Page 9 line 333: this section seem to relate to new results and probably doesn't belong in a review. 

The discussion would probably benefit of other examples of virus imaging taken on other beamline around the world and comparison of various resolution achieved.  

Author Response

Dear Editor,

 Please find in your online submission system our manuscript entitled “IMAGING OF VIRUS-INFECTED CELLS WITH SOFT X-RAY TOMOGRAPHY AT MISTRAL BEAMLINE” which we would like to publish in Viruses . Here, we have provided a point-by-point response to the reviewer 3’s comments:

Page 2 line 88: one more beamline offers SXT, the XM-2 beamline at Lawrence Berkeley National Laboratory

We thank the reviewer for the suggestion. Beamline XM-2 of the Advanced Light Source at Lawrence Berkeley National Laboratory, also referred to as beamline 2.1, was already included in the list of SXT beamlines provided in page 2, although the synchrotron name had an error. To avoid any confusion, we have included both beamline names in the text:

Cryo-SXT is available at HZB X-Ray Microscopy Beamline U41-PGM1-XM at BESSY II (Berlin, Germany), 2.1 beamline (XM-2) at the Advanced Light Source (Berkeley, USA), B24 beamline at Diamond Light Source (Oxford, UK) and BL09-MISTRAL beamline at ALBA synchrotron (Barcelona, Spain).

References to the articles describing each of these beamlines have also been added.

Page 3 line 111: it would be nice to explain why one would choose partial versus full tilt and which beamlines around the world offers what type of tilt.

The manuscript has been updated to include reviewer’s suggestion. The text now reads:

“The achieved resolution is also dependent on the chosen data-collection scheme (i.e. if full or partial tilt series are performed) and the achieved signal-to-noise values. The choice of data-collection scheme depends on the sample holder: capillary tubes, such as those used in XM-2 beamline, can be rotated 360 degrees (full tilt), while microscopy grids used in the rest of beamlines can only be tilted for ~140 degrees (partial tilt). On the other hand, signal-to-noise values will depend on the thickness of the biological samples, for which 10 μm is usually considered as the maximum compatible.”

Page 3 line 127: Samples can also be housed in capillary tubes for full tilt rotation since the EM grids support doesn't allow for full tilt.

The reviewer is correct. We have added a sentence in the manuscript to include a mention to this possibility. The updated text now reads:

The experimental workflow for cryo-SXT methodology starts with the seeding or deposition of cells on carbon-coated microscopy grids, previously sterilized by UV irradiation (Figure 1a). Alternatively, cells that grow in solution can also be housed in thin-walled glass capillary tubes, which allow for specimen to be rotated through 360° [McDermott et al. 2009]. Grids used for cryo-SXT have a mesh dividing the grid area into different regions, each of which are identified by an alphanumerical code that facilitates identification of the regions of interest during data collection.”

The new reference is: McDermott,G; Le Gros,MA; Knoechel,CG; Uchida, M; Larabell, CA. Soft X-ray tomography and cryogenic light microscopy: the cool combination in cellular imaging. Trends in Cell Biology. 2009, 19 (11): 587-595. doi: 10.1016/j.tcb.2009.08.005.

Page 4 line 151: a short discussion of the potential effect of fixation on samples that are supposed to be in their native state (cryofixation) would be nice as well as a comparison in this case of sample fixed for block face imaging which would lead to similar resolution.

We have updated the text to include the potential effects of sample fixation as well as a description of the vitrification process and its relevance. The text reads:

“Finally, non-infectious, inactivated samples are washed out and grids holding the samples are cryopreserved. In conventional EM fixation, samples are prepared by chemical fixation with osmium tetroxide or aldehydes at room temperature. Then, the sample is dehydrated with organic solvents, infiltrated and embedded in epoxy resin. This process induces artefacts such as distortion of the cell morphology, which makes it inadequate for preserving membrane-rich structures and can led to misinterpretations of cellular structures [33,34]. A much better alternative for preservation of cellular structures is cryofixation of the specimen, either by high-pressure freezing (HPF), which combines jets of liquid nitrogen with very high pressures, or ultra-rapid cryofixation by plunge freezing, by rapid immersion of the grid in a cryogen. Both techniques obtain vitrified biological samples avoiding the formation of ice crystals which could destroy the sample, or containing ice crystals small enough not to damage the ultrastructure of the cells [35]. During vitrification, sample is frozen at a very high cooling rate, instantly fixing the sample in an amorphous, glass-like phase, thus keeping all cellular components in a close-to-life state.

For thin samples of less than 20 μm thickness, as is the case for cryo-SXT studies, plunge freezing vitrification is nowadays the method of choice. There, the excess of so-lution on the grid holding the sample is removed by blotting, right before the grid is rapidly plunged into liquid ethane. Plunge-freezing vitrification of cryo-SXT samples is relatively straightforward, especially in comparison to sample preparation for cryo-EM, since cryo-SXT is less sensitive to devitrification processes affecting areas smaller than ~10×10 nm2. In general, when plunge-freezing cryo-SXT samples, the most critical parameter that needs to be optimized is the blotting time.”

Regarding the comparison with sample preparation for block face imaging, a new comment on this topic has been included in the Discussion section. The new text reads:

“[…] FIB-SEM is a powerful technique for 3D imaging of sub-micron features in a sample, in which the specimen is sequentially milled away using an ion beam while the newly exposed surface is imaged using an electron beam, yielding reconstructed volumes with resolutions <10 nm in all directions [68]. However, this process is slow and requires sample to be fixated in a block, which can led to significant sample distortions [34]. To bypass the need for chemical fixation, cryo-FIB-SEM has recently been developed to image vitrified biological samples at cryogenic temperature. Nevertheless, further studies are still required before cryo-FIB-SEM can be established as a standard method for whole cell imaging [69].”

Page 4 line 152: reference 28 seems inadequate, this reviewer cannot find any reference to 2 % formaldehyde in this citation.

We thank the reviewer for spotting this error. In the updated version of the manuscript, this reference has been changed for the correct one, which does discuss the effects of formaldehyde treatment on virus-infected cells:

Möller, L; Schünadel, L; Nitsche, A; Schwebke, I; Hanisch, M; Laue, M. Evaluation of Virus Inactivation by Formaldehyde to Enhance Biosafety of Diagnostic Electron Microscopy. Viruses. 2015, 7(2): 666–679. doi: 10.3390/v7020666.

Page 4 line 160: adding the vitrification process would improve the manuscript since the vitrification is the stepping stone for sample quality.

We do agree with the reviewer in the importance of proper vitrification of the sample. A description of the vitrification process and its relevance has been included in the text. See the response for comment on Page 4, line 150 for more details.

Page 4 figure 1: this figure seems to be a little confusing, at least for panel c to e with reduced image size where no real details can be seen. A schematic representation of the workflow would work better.

We thank the reviewer for pointing the lack of clarity in Figure 1. We have updated the figure to make it easier for the reader to follow the workflow. In particular, we have removed panel e) from the figure and modified panel c) to make it easier to understand: we have highlighted the sample holder in the loading station, added a schematic drawing of a loaded sample holder next to the picture and indicated where the loaded grid sits. We still included panel d) in the figure, as this is used to illustrate that the samples are loaded into the microscope at that stage of the workflow.

Page 5 line 194: the LBNL beamline has a 360 degres stage.

The text has been updated to include this information. The manuscript now reads:

The typical angular range for these tomograms is ±70 or ±60 degrees for samples on grids, or up to 360 degrees for samples in capillary tubes.

Page 6 line 247: "figfigffure" should be replaced by "fig".

We thank the reviewer for spotting the typo in the pdf version; this is not present in the updated text.

Page 9 line 333: this section seems to relate to new results and probably doesn't belong in a review.

The reviewer is correct in that results presented in section 3.3 have not yet been published elsewhere. However, we think that these results provide a valuable contribution to this review, since they illustrate the kind of information that can be extracted from aligned tomograms, representing the “raw” results (i.e. prior to segmentation analysis) that can be obtained by cryo-SXT.

The discussion would probably benefit of other examples of virus imaging taken on other beamline around the world and comparison of various resolution achieved.

Most of the recent examples of virus-related studies performed at soft X-ray microscopy beamlines other than MISTRAL are cited at the beginning of section 3; for the sake of completeness, another citation has been added to that list in the updated manuscript:

Hagen, C.; Guttmann, P.; Klupp, B.; Werner, S.; Rehbein, S.; Mettenleiter, T.C.; Schneider, G.; Grünewald, K. Correlative VIS-fluorescence and soft X-ray cryo-microscopy/tomography of adherent cells. J. Struct. Biol. 2012, 177: 193–201. doi: https://doi.org/10.1016/
j.jsb.2011.12.012.

Also, several of these examples were already further described in the Discussion section.

Regarding the resolution achieved in these studies, it is difficult to compare for all of them, since not all of these articles describe the achieved resolution in a comparable way. Instead, we have focused on the resolutions that can be achieved at each of the cryo-SXT beamlines and included this comparison in the manuscript. The new text now reads:

“The resolution of the obtained cryo-SXT tomograms mainly depends on the zone plate (ZP) installed in the beamline optics. The ZP basically acts as an “objective lens”, gathering the photon beam transmitted through the sample and focusing the magnified image of the specimen onto the detector. The resolving power is proportional to the smallest outermost zone width of the zone plate, which defines the higher achievable resolution limit. Each cryo-SXT beamline has chosen the specifications of their ZP as a compromise between resolution, field of view and depth of field: the smaller the width of the ZP outermost zone, the higher the achieved resolution, but also the smaller the asso-ciated field of view and depth of field. HZB-PGM1-XM beamline (BESSY II) is equipped with a 25 nm-outermost zone width ZPs [25]; both B24 beamline (DLS synchrotron) and MISTRAL beamline (ALBA synchrotron) use 25 nm and a 40 nm-outermost zone width ZPs [47,70,71]; and XM-2 beamline (Advanced Light Source) is equipped with 50 nm and 80 nm-outermost zone width ZPs [26].”
